# RANDOM EFFECT BANDITS USING H-LIKELIHOOD PROCEDURE

## ABSTRACT

Stochastic multi-armed bandit (SMAB) is a fundamental framework for sequential decision-making in reinforcement learning, where an agent must balance exploration and exploitation to maximize cumulative rewards. Recently, random effect SMAB has been proposed where reward feedback is modeled as random effect. However, it has not been well formulated yet in likelihood perspectives. Furthermore, individual noise variance can be arm-dependent. We propose a novel random effect upper confidence bound (ReUCBHL) algorithm, based on h-likelihood. The likelihood approach is conceptually easy and can be implemented by simply minimizing the loss (negative h-likelihood). The algorithm can be applied to SMAB with univariate and multivariate rewards under arm-dependent noise variances. It can be further extended to contextual multivariate bandit. Theoretical justification and simulation studies demonstrate that ReUCBHL consistently achieves better regret performance compared to the baseline algorithms. These results highlight the effectiveness of the proposed algorithm.

## 1 INTRODUCTION

Many real-world decision-making problems involve actions whose outcomes share a common baseline, while their variability differs substantially across actions. In health care, for instance, treatments for the same condition may be influenced by a common patient-level effect, yet individual treatments exhibit different levels of uncertainty due to trial size or biological variability. Recently, Ghosh et al. (2024) applied reBandit algorithm in a mobile health intervention to reduce cannabis use among emerging adults, where random-effects modeling was leveraged to share information across individuals. Similar structures arise in personalized education, where student performance on different test items depends on a shared latent skill but exhibits varying levels of measurement noise. Similarly, in recommendation systems, where global popularity trends influence all items while feedback for niche products is considerably noisier than for widely adopted ones.

Bandit framework is widely used in reinforcement learning to model the trade-off between exploration and exploitation. The most popular one is the stochastic multi-armed bandit (SMAB) (Lai & Robbins, 1985; Auer et al., 2002; Zhu & Kveton, 2022a) where at each time step the agent sequentially selects an arm/action to maximize the total accumulated rewards over $n$ rounds of play. Each arm generates random rewards from unknown reward distribution. Objective of SMABs is to minimize the total regret. Through experience, the agent faces trade-off between exploration (trying new actions which might give higher reward in future) and exploitation (drawing the arm with maximum reward in past). SMABs have been analyzed using either regression (fixed effect) models or Bayesian models. In SMABs, the upper confidence bound (UCB) algorithm is the most popular due to its simplicity of implementation and established results on regret bound (Lai & Robbins, 1985; Auer et al., 2002; Dani et al., 2008; Abbasi-Yadkori et al., 2011; Zhu & Kveton, 2022a). Many versions of UCB algorithms have been developed under fixed effect models which operate on confidence bounds (Auer et al., 2002; Dani et al., 2008; Abbasi-Yadkori et al., 2011; Garivier & Cappe, 2011). Thompson sampling (TS) algorithm is the most popular due to theoretical advantages with good performance (Aggarwal & Navin, 2012; 2013; Russo & Roy, 2016; Abeille & Lazaric, 2017; Aouali et al., 2023). Kaufmann et al. (2012) introduced Bayes upper confidence bound (BUCB) algorithm.

While the prior is a blessing when correctly specified, a misspecified prior could be a curse (Zhu & Kveton, 2022a). However, the prior is hardly checkable via data. Zhu & Kveton (2022a) proposed the use of a random effect model for SMABs. They proposed random effect UCB (ReUCB) algorithm. They showed that ReUCB algorithm performs much better than UCB and can be even better than TS. Rewards from each arm can be multi-dimensional, so Lee et al. (2024) studied contextual SMABs with multiple rewards from each arm. Recently, random effect models have been of interest for subject-specific predictions in statistical literature (Lee & Nelder, 1996). Distribution of random effects could be checkable and various model checking procedures have been developed (Lee et al., 2016). Lee & Nelder (1996) introduced the h-likelihood for inference from the model with additional random parameters. However, their h-likelihood may not give optimal estimation. Various alternatives have been developed to estimate parameters. Existing state-of-art algorithms have used different procedures to estimate various parameters in random effect models. For example, the best linear unbiased predictors (BLUPs) for random parameters, maximum likelihood estimators (MLEs), weighted least squares (WLS) estimators for fixed effects, method of moment (MM) and expectation-maximization (EM) for variance parameters. It has long been recognized that noise variance could be arm-dependent. However, difficulty in implementing efficient estimation algorithm prevents full development of UCB algorithm for random effect model approach. Recently, Lee & Lee (2023) defined the new h-likelihood for random effect deep neural network models, whose simple maximization provides an optimal estimation of all fixed and random parameters (Lee & Lee, 2025). In this study, we extend Lee & Lee (2023)'s h-likelihood to arm-dependent SMABs to develop the random effect ReUCB (ReUCBHL) algorithm. Lee et al. (2024) developed UCB algorithm for random effect contextual SMAB with multi-dimensional reward. Our algorithm can be easily extended to improve their contextual SMABs. An immediate advantage of our approach is that it is straightforwardly implemented by simple minimization of the loss (negative h-likelihood) function.

## 2 FORMULATION OF BANDIT MODEL

We consider the SMAB with $K$ arms, where each arm $k \in [K] = \{1, 2, ..., K\}$ generates i.i.d. random reward $r_{k,t}$ at the round $t \in [n] = \{1, 2, ..., n\}$.

### 2.1 FIXED EFFECT SMAB

At the $t$th round, the reward $r_{k,t}$ of the arm $k$ is generated from the fixed effect SMAB:

$$r_{k,t} = \mu_k + e_{k,t} \tag{1}$$

where $r_{k,t}$ is the random reward generated from the arm $k$ in t-th pull, $\mu_k$ is the fixed unknown mean reward of arm $k$ and $e_{k,t} \sim \mathcal{N}(0, \sigma^2)$ is the noise term. Auer et al. (2002) introduced the UCB algorithm under the regression model (1). Bayesian models have been introduced by allowing prior distributions for $\mu_k$ and $\sigma^2$. TS (Aggarwal & Navin, 2012; 2013) and BUCB (Kaufmann et al., 2012) algorithms have been developed based on posterior of rewards.

In real-world applications, however, the noise variance of reward could be arm-dependent $e_{k,t} \sim \mathcal{N}(0, \sigma_k^2)$ (Kirschner & Krause, 2018). Simultaneous fitting algorithm for fixed means $\mu_k$ and variances $\sigma_k^2$ have been developed for analysis of quality improvement experiments (Lee & Nelder, 1998). SMAB with arm-dependent noise variance was introduced by Cowan et al. (2018). Kirschner & Krause (2018) proposed a weighted least squares method to estimate the unknown reward function by assuming that the variance of the noise at each round $t$ is a function of the chosen action. Zhao et al. (2022) considered SMABs where the unknown reward function belongs to a more general class of functions.

### 2.2 RANDOM EFFECT SMAB WITH ARM-INDEPENDENT NOISE

When the number of arms is large, the prediction of rewards based on the fixed effect SMAB could be unreliable. For a better prediction of rewards, Zhu & Kveton (2022a) proposed the random effect SMAB:

$$r_{k,t} = \mu_k + e_{k,t} \text{ with } \mu_k = \mu_0 + \delta_k, \tag{2}$$

where $\mu_0$ is the fixed common mean of arms and $\delta_k \sim \mathcal{N}(0, \sigma_0^2)$ are random effects. In this model, $\mu_k$ is the random mean reward of the arm $k$ and the noise variance is arm-independent, $var(e_{k,t}) = \sigma^2$. For estimation, Zhu & Kveton (2022a) used the BLUP for $\delta_k$, genearalized least square estimator for $\mu_0$ and method of moments for fixed variance parameters $\sigma_0^2$ and $\sigma^2$. Based on these estimates, they developed ReUCB algorithm. They derived an upper bound on $n$-round regret for this algorithm and empirically showed that ReUCB can outperform TS algorithm.

## 2.3 RANDOM EFFECT SMAB WITH ARM-DEPENDENT NOISE

In this study, we consider the arm-dependent random effect SMAB:
$$r_{k,t} = \mu_k + e_{k,t} \text{ with } \mu_k = \mu_0 + \delta_k, \tag{3}$$
where $\mu_0$ and $\delta_k \sim \mathcal{N}(0, \sigma_0^2)$ are the same as the model (2), but $e_{k,t} \sim \mathcal{N}(0, \sigma_k^2)$. This model leads to the within arm-dependent variance $var(r_{k,t}|\delta_k) = \sigma_k^2$ and between arm-independent variance $var(\delta_k) = var(\mu_k) = \sigma_0^2$ to give the total arm-dependent variance $var(r_{k,t}) = \sigma_k^2 + \sigma_0^2$.

Zhu & Kveton (2022a) considered arm-dependent random effect SMAB (3) and used the method of moment to estimate $\sigma_k^2$. However, they used the BLUP procedure to predict random rewards under the arm-independent random effect SMAB (2) by taking $\sigma_*^2 = \max \sigma_k^2$ as the noise variance of all the arms. In this study, we develop an algorithm for the arm-dependent random effect SMAB (3) by simply maximizing h-likelihood, and show its advantages.

Suppose that we observe the multi-dimensional reward vector $\mathbf{r} = (r_{A_1,1}, ..., r_{A_n,n})^T$ over $n$ rounds of play, where $A_t \in [K]$ is the chosen arm in the round $t$. Then, the multi-dimensional arm-dependent SMAB (3) can be expressed by
$$\mathbf{r} = \mathbf{1}\mu_0 + \mathbf{Z}\delta + \mathbf{e} \tag{4}$$
where $\mathbf{1}$ is the column vector with all elements 1, $\mathbf{Z}$ is the $n \times K$ matrix whose $(t, A_t)$th elements are 1 for $t = 1, ..., n$ and the rest are 0, $\delta = (\delta_1, ..., \delta_K)^T \sim MVN(\mathbf{0}, \mathbf{D}_K)$ with $\mathbf{D}_K = \sigma_0^2 \mathbf{I}_K$, $\mathbf{e} = (e_{A_1,1}, ..., e_{A_n,n})^T \sim MVN(\mathbf{0}, \mathbf{\Sigma}_n)$ and $\Sigma_n$ is the $n$-dimensional diagonal matrix whose $t$th element is $\sigma_{A_t}^2$.

# 3 RANDOM EFFECT SMAB WITH ARM-DEPENDENT NOISE

## 3.1 H-LIKELIHOOD

Lee & Lee (2023) derived the h-likelihood, applicable to arm-independent SMAB (2). Under the arm-dependent random effect SMABs (3) and (4), let the $\delta^c = \mathbf{B}^{1/2}\delta$ where $\mathbf{B} = (\mathbf{Z}^T\mathbf{\Sigma}^{-1}\mathbf{Z} + \mathbf{D}_K^{-1})$ and $\mathbf{B}^{1/2}$ is computed by Cholesky decomposition. Then, given observed reward $\mathbf{r}$, the h-likelihood for the random effect $\delta^c$ and fixed parameters $\theta = (\mu_0, \sigma_0^2, \sigma_1^2, ..., \sigma_K^2)^T$ in the $n$ round of play can be defined by

$$
\begin{aligned}
h(\theta, \delta^c; \mathbf{r}) = & -\frac{1}{2}(\mathbf{r} - \mathbf{1}\mu_0 - \mathbf{Z}\mathbf{B}^{1/2}\delta^c)^T \Sigma^{-1}(\mathbf{r} - \mathbf{1}\mu_0 - \mathbf{Z}\mathbf{B}^{1/2}\delta^c) - \frac{1}{2}\log(2\pi\mathbf{\Sigma}) \\
& -\frac{1}{2}\delta^{cT}\mathbf{B}^{1/2}\mathbf{D}_K^{-1}\mathbf{B}^{1/2}\delta^c - \frac{1}{2}\log(2\pi\mathbf{B}^{1/2}\mathbf{D}_K\mathbf{B}^{1/2}).
\end{aligned}
$$

The simple joint maximization of the h-likelihood gives the MLEs for all fixed parameters $\theta$ and the BLUPs, $\widehat{\delta} = E(\delta|\mathbf{r})|_{\theta=\hat{\theta}}$ for random effects $\delta$. There is no need to develop different estimation procedures for the fixed effect $\mu_0$, and dispersion parameters $\sigma_0^2, \sigma_1^2, ..., \sigma_K^2$ and random effects $\delta$.

## 3.2 REUCBHL ALGORITHM

We propose a UCB algorithm for arm-dependent random-effect SMABs using h-likelihood. UCB algorithm works by associating an upper confidence index to each arm and pulling the arm with the highest index value. The upper index is the sum of mean reward estimate and a weighted standard deviation of that estimate. The proposed ReUCBHL algorithm is given in Table 1. ReUCBHL is initialized by pulling each arm once. The upper confidence index of arm $k$ in round $t$ is calculated as
$$U_{k,t} = \hat{\mu}_{k,t} + \widehat{c}_{k,t} \tag{5}$$

where $\hat{\mu}_{k,t} = \widehat{\mu}_{0,t} + \widehat{\delta}_{k,t}$, $\widehat{c}_{k,t} = \sqrt{a\widehat{\tau}_{k,t}^2 \log(t)}$ is the uncertainty bonus, $\widehat{\tau}_{k,t}^2 = \widehat{var}(\hat{\mu}_{k,t})$ and $a > 0$ is a tuning parameter. In each round $t$, ReUCBHL pulls the arm with the highest index value. If two or more arms have same highest value, randomly pull one of the arms.

| | |
|---|---|
| 1: | Pull each arm once |
| 2: | **for** each round $t = k+1, 2, ..., n$ **do** |
| 3: | **for** $k = 1, 2, ..., K$ **do** |
| 4: | $U_{k,t} \leftarrow \hat{\mu}_{k,t} + \widehat{c}_{k,t}$ |
| 5: | **end for** |
| 6: | Pull the arm $A_t = \text{argmax}_{k \in [K]} U_{k,t}$ |
| 7: | Observe the reward $r_{A_t, n_{A_t}}$ |
| 8: | Update all statistics |
| 9: | **end for** |

Table 1: ReUCBHL algorithm for SMAB

We derive an upper bound on the $n$-round regret of ReUCBHL algorithm given in Table 1. Under the SMAB model (3), $\mu_k$ are random variables. Assuming that $r_{k,j} \sim \mathcal{N}(\mu_k, \sigma_k^2)$, $\mu_k \sim \mathcal{N}(\mu_0, \sigma_0^2)$ and $(\hat{\mu}_{k,t} - \mu_k)|\delta_k \sim \mathcal{N}((w_k - 1)\delta_k, w_k^2 \sigma_k^2/n_k)$. We introduce a new notion of regret, motivated by h-likelihood inference in random-effects model. Once the random effects $\delta = (\delta_1, \delta_2, \ldots, \delta_K)^T$ are generated from some distribution, then they are realized as fixed $\delta_0 = (\delta_{01}, \delta_{02}, \ldots, \delta_{0K})^T$. Let $A_t$ be the arm pulled at round $t$ and $A^* = \text{argmax}_{k \in [K]}(\mu_0 + \delta_{0k})$ is the optimal arm under the realized values $\delta_0$. In this study, we define regret of a bandit algorithm under random effect model (3) after n rounds as

$$R_n = E\left\{\sum_{t=1}^{n} (\mu_{A^*} - \mu_{A_t,t})\right\},$$

where $\mu_{A^*} = \max_{k \in [K]}(\mu_0 + \delta_{0k})$ is a fixed unknown constant given the realized values $\delta_0$, $A_t = \arg\max_{k \in [K]}(U_{k,t})$ in (5) and the expectation is over randomness in reward.

Unlike Bayes regret (Zhu & Kveton, 2022a), this definition does not average over the prior distribution of $\mu = (\mu_1, \ldots, \mu_K)$. Instead, it treats them as realized values of unobservable random variables. The maximum h-likelihood estimator $\hat{\mu}_{A_t,t}$ is the optimal estimator of the realized value $\mu_{A_t,t}$, by achieving generalized Cramer-Rao lower bound (Lee & Lee, 2025).

**Theorem 1.** *Under the SMAB (3) with arm-dependent noise, the $n$-round regret of ReUCBHL is*

$$R_n \leq C\sqrt{\log n} \frac{\sqrt{\sum_{k=1}^{K} \sigma_k^2}}{\sigma_0} \sqrt{\sigma_0^2 n + \sum_{k=1}^{K} \sigma_k^2} + O(\delta_{\max} \sum_{k=1}^{K} (\sigma_k^2/\sigma_0^2) \log n) + O(K\sqrt{\log n}),$$

where $\delta_{\max} = \max_{k \in [K]}\{\delta_{0k}\}$ and constant $C = 4\sqrt{2}$.

## 3.3 RELATED WORKS

- UCB (Auer et al., 2002): UCB algorithm is developed under the fixed effect SMAB (1) wherein the agent pulls the arm with the highest upper confidence index, using the MLEs.

- BUCB (Kaufmann et al., 2012): Bayesian models assumes a prior distribution on the fixed parameters $\theta$. BUCB algorithm has been proposed using quantiles of posterior distribution.

- TS (Aggarwal & Navin, 2012; Russo et al., 2018; Zhu & Kveton, 2022a): TS algorithm chooses the arm with highest expected reward under the posterior distribution.

- ReUCB1 (Zhu & Kveton, 2022a): ReUCB1 stands for ReUCB algorithm under arm-independent random effect SMAB (2).

- ReUCB2 (Zhu & Kveton, 2022a): ReUCB2 is for arm-dependent random effect SMAB. However, they used the maximum noise variance as the common noise variance and used the estimation procedure under the arm-independent random effect SMAB (2). Thus, they do not fully exploit the advantage of arm-dependent random effect SMAB (3).

- ReUCBHL (the proposed algorithm): ReUCBHL is an extension of ReUCB algorithm to arm-dependent random effect SMAB (3). In this study, ReUCBHL stands for the use of h-likelihood algorithm under arm-dependent random effect SMAB (3).

As ReUCBHL uses an upper confidence index, it is similar to UCB algorithm. The difference lies in the fact that UCB assumes the mean of each arm as fixed and uses MLEs, whereas ReUCBHL assumes the mean as random and uses h-likelihood estimates, which are the MLEs for fixed parameters $\theta$ and the BLUPs for random effects. ReUCBHL is closely related to ReUCB because they are based on random effect models. The difference is that ReUCB1 and ReUCB2 use the method of moments for variance estimators, the generalized least square estimator for $\mu_0$ and the BLUPs for random effects, whereas ReUCBHL use maximum h-likelihood estimators for all fixed and random parameters. Parameter estimation methods provide similar result because ReUCB1 and ReUCBHL under arm-independent random effect SMAB (2) provide almost identical results. ReUCBHL gives BLUP estimators for the arm-dependent random effect SMAB. The numerical study shows that ReUCBHL performs generally the best.

## 3.4 Numerical studies for synthetic experiments

For random effect SMAB (3), we first set $\mu_k \sim \mathcal{N}(1, 0.04)$ with $K = 50$ and $n = 10,000$. Second, $\mu_k$ are drawn from uniform distribution $\mathcal{U}(1, 2)$. Then, we compare the performance of ReUCBHL in terms of cumulative regret over $n$ rounds of play with four other benchmark algorithms UCB (Auer et al., 2002), TS (Zhu & Kveton, 2022a), BUCB (Kaufmann et al., 2012) and ReUCB (Zhu & Kveton, 2022a). ReUCBs with arm-independent and arm-dependent SMAB are denoted as ReUCB1 and ReUCB2, respectively. Each experiment is based on 1,000 independent simulation runs. We consider arm-independent and arm-dependent cases to study the performance of various algorithms. ReUCBHL is generally the best among algorithms.

i) Arm-independent case: We assume $\sigma_k^2 = \sigma^2 = 0.25$ for all arms. Zhu & Kveton (2022a) noted that the Gaussian random-effect model is robust against the misspecification for distribution of random effects. Figure 1(a) for $\mu_k \sim \mathcal{N}(1, 0.04)$ and Figure 1(b) for $\mu_k \sim \mathcal{U}(1, 2)$ show plots of cumulative regret as a function of time horizon. Algorithms perform similarly under normal and uniform assumptions. The performance of TS is better than UCB and BUCB but worse than ReUCB1, ReUCB2 and ReUCBHL. In arm-independent cases, the ReUCB1 should be the best. ReUCBHL and ReUCB2 are almost identical to the ReUCB1 in arm-independent cases.

ii) Arm-dependent cases: We generate noise variances as $\sigma_k^2 = 0.25 \times k$. The regret performance of the algorithms is shown in Figure 1(c) for $\mu_k \sim \mathcal{N}(1, 0.04)$ and Figure 1(d) for $\mu_k \sim \mathcal{U}(1, 2)$. We observe that ReUCBHL outperforms the other algorithms. ReUCB1 works poor under arm-dependent cases. Thus, the arm-dependent random effect SMAB is preferred to the arm-independent random effect SMAB.

## 3.5 Experiments on real data

We apply SMAB algorithm to recommendation problem. We consider MoiveLens dataset (Lam & Herlocker, 2016), which contains almost 1 million ratings, 4,000 users and 6,000 movies. Our goal is to identify the movie with highest rating for a specific user group. We preprocess the data following the Katariya et al. (2017). Our learning problem is formulated as follows. Define a user group for every unique combination of gender, age group and occupation. The total number of user groups is 241. For each user group and movie pair, we average the ratings of all the users in that group that rate the movie and learn a low-rank approximation to the underlying rating matrix $M$ using the algorithm in Keshavan et al. (2010). The algorithm automatically detects the rank of the matrix to be 5. We randomly choose $J = 128$ user groups and $K = 128$ movies. The reward for choosing user group $j \in [J]$ and movie $k \in [K]$ is a categorical random variable over five-star ratings. We estimate its parameters based on the assumption that the ratings are normally distributed with a fixed variance, conditioned on the completed ratings. Our results are averaged over 200 runs. In each run, user $j$ is chosen uniformly at random from [128] and it represents a bandit instance in that run. The goal is to learn the most rewarding movie for the user $j$. We model this problem as a

random-effect SMAB with $K = 128$ arms, one per movie, where the mean reward of movie $k$ by user $j$ is the $(j, k)$th element $M_{j,k}$ of $M$.

i) Arm-independent case: Following Zhu & Kveton (2022a), the rewards are generated from $\mathcal{N}(M_{j,k}, 0.796^2)$. Figure 1(e) shows that TS performs better than UCB and BUCB but worse than ReUCB1, REUCB2 and ReUCBHL.

ii) Arm-dependent case: The rewards are generated from $\mathcal{N}(M_{j,k}, 0.796^2 \times \log(k+1))$. The regret performance of the algorithms is shown in Figure 1(f). We observe that ReUCBHL outperforms the other algorithms. ReUCB1 works poor in arm-dependent case. Thus, arm-dependent assumption enhances the performance and ReUCBHL works well in both arm-independent and arm-dependent cases.

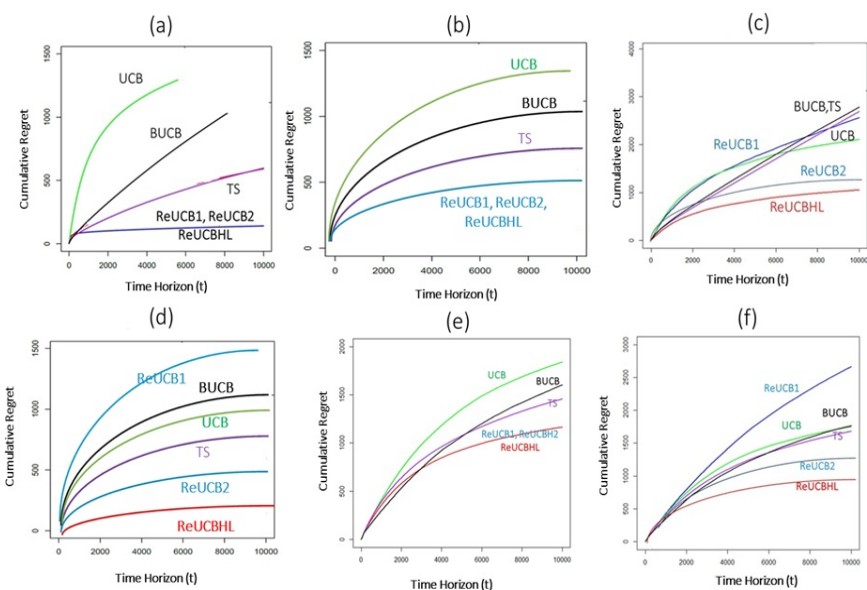

Figure 1: Comparison of the average cumulative regrets (a) $\mu_k \sim \mathcal{N}(1, 0.04)$, (b) $\mu_k \sim \mathcal{U}(1, 2)$ arm-independent noise, (c) $\mu_k \sim \mathcal{N}(1, 0.04)$, (d) $\mu_k \sim \mathcal{U}(1, 2)$ arm-dependent noise in numerical studies for synthetic experiments and (e) arm-independent noise, (f) arm-dependent noise in experiments on real data under the random effect SMAB.

# 4 MULTIVARIATE SMAB

Lee et al. (2024) introduced a new variant of contextual SMAB where the reward is formulated by multi-dimensional random effect SMAB. The correlations among multiple rewards arise due to the sharing of stochastic coefficients called random effects. To address this setting, they proposed mixed effect contextual UCB (ME-CUCB) algorithm for contextual SMAB with arm independent noise. In this section, we investigate how our algorithm can be extended to enhance the performance of their multi-dimensional contextual SMAB framework.

## 4.1 MULTI-DIMENSIONAL CONTEXTUAL RANDOM EFFECT SMAB WITH ARM-DEPENDENT NOISE

In the contextual stochastic multi-armed bandit (contextual SMAB) framework, at each round $t \in [n]$, the learner observes a context vector $\mathbf{x}_t$, pulls an arm $k \in [K]$ conditioned on the context and receives a random reward. The contextual SMAB can generate multi-dimensional rewards from each arm. Recently, contextual multi-dimensional SMABs have attracted increasing interest. We consider a multi-dimensional SMAB with K arms, where pulling an arm $k \in [K]$ generates an $m$-dimensional column vector of rewards $\mathbf{r}_{k,t}$ at the round $t \in [n]$. The multivariate reward $\mathbf{r}_{k,t}$ can be

correlated at each time point but are independent across time. In this section, we focus on contextual random effect SMAB with arm-dependent noise:

$$\mathbf{r}_{k,t} = \mu_{k,t} + \mathbf{e}_{k,t} \ \text{ with } \ \mu_{k,t} = \mathbf{X}_{k,t}\beta + \mathbf{Z}_{k,t}\delta_k \tag{6}$$

where the $m \times p$ matrix $\mathbf{X}_{k,t}$ and the $m \times q$ matrix $\mathbf{Z}_{k,t}$ are the context matrices for the $p$-dimensional fixed effect $\beta$ and the $q$-dimensional random effect $\delta_k \sim \mathbf{N}(\mathbf{0}, \mathbf{\Sigma}_0)$ with $q \times q$ covariance matrix $\mathbf{\Sigma}_0$, respectively. Both $\mathbf{X}_{k,t}$ and $\mathbf{Z}_{k,t}$ may vary over time $t$ and $\mathbf{e}_{k,t} \sim N(0, \sigma_k^2 \mathbf{I}_m)$ is the multi-dimensional noise and $\mu_{k,t}$ is the multi-dimensional random mean reward. Lee et al. (2024) studied random effect SMAB with arm-independent noise, $\sigma_1^2 = \cdots = \sigma_K^2 = \sigma^2$.

Lee et al. (2024) used the weighted least squares estimators for fixed effect, the BLUPs for estimating random effects and the EM algorithm for estimating variances. In the h-likelihood approach, the simple joint maximization gives the MLEs for all fixed parameters $\theta$ and the BLUPs, $\widehat{\delta} = E(\delta|\mathbf{r})|_{\theta=\hat{\theta}}$, for random effects. Furthermore, the h-likelihood algorithm in Section 3.1 is straightforwardly extended to multi-dimensional contextual SMABs by simply replacing $\mathbf{1}\mu_0$ by $\mathbf{X}\beta$ and $\sigma_0^2$ by $\mathbf{\Sigma}_0$.

## 4.2 H-LIKELIHOOD

Under the model (6), let the $\delta^c = \mathbf{B}^{1/2}\delta$ where $\mathbf{B} = (\mathbf{Z}^T \mathbf{\Sigma}^{-1} \mathbf{Z} + \mathbf{D}^{-1})$. Given $mn$-dimensional column vector the observed reward $\mathbf{r} = (\mathbf{r}_{A_1,1}^T, ..., \mathbf{r}_{A_n,n}^T)^T$, the h-likelihood for fixed parameters $\theta = (\beta, \mathbf{\Sigma}_0, \sigma_1^2, ..., \sigma_K^2)$ and the random effect $\delta^c$ in the $n$ round of play can be defined as

$$
\begin{aligned}
h(\theta, \delta^c; \mathbf{r}) &= -\frac{1}{2}(\mathbf{r} - \mathbf{X}\beta - \mathbf{Z}\mathbf{B}^{-1/2}\delta^c)^T \mathbf{\Sigma}^{-1}(\mathbf{r} - \mathbf{X}\beta - \mathbf{Z}\mathbf{B}^{-1/2}\delta^c) - \frac{1}{2}\log(2\pi\mathbf{\Sigma}) \\
&\quad - \frac{1}{2}\delta^{cT}\mathbf{B}^{-1/2}\mathbf{D}^{-1}\mathbf{B}^{-1/2}\delta^c - \frac{1}{2}\log(2\pi\mathbf{B}^{1/2}\mathbf{D}\mathbf{B}^{1/2}),
\end{aligned}
$$

where $\mathbf{X} = (\mathbf{X}_{A_1,1}^T, ..., \mathbf{X}_{A_n,n}^T)^T$ is the $mn \times p$ context matrix for $\beta$, $\mathbf{Z} = (\mathbf{Z}_{A_1,1}^T \otimes \mathbf{a}_{A_1}^T, ..., \mathbf{Z}_{A_n,n}^T \otimes \mathbf{a}_{A_n}^T)^T$ is the $mn \times qK$ context matrix for $\delta$, where $\mathbf{a}_{A_t}$ is the $q$-dimensional column vector with value 1 for $A_t$th element and 0 for otherwise. The $qK$ dimensional random effects $\delta = (\delta_1^T, ..., \delta_K^T)^T \sim MVN(\mathbf{0}, \mathbf{D})$ with the $qK \times qK$ matrix $\mathbf{D} = \mathbf{\Sigma}_0 \otimes \mathbf{I}_K$, the $mn$ dimensional noise $\mathbf{e} = (\mathbf{e}_{A_1,1}^T, ..., \mathbf{e}_{A_n,n}^T)^T \sim MVN(\mathbf{0}, \mathbf{\Sigma})$ and the $mn \times mn$ matrix $\mathbf{\Sigma} = \mathbf{I}_m \otimes \mathbf{diag}\{\sigma_{A_1}^2, ..., \sigma_{A_n}^2\}$. Here $\otimes$ denotes Kronecker product. The simple joint maximization gives the MLEs for all fixed parameters $\theta$ and the BLUPs, $\widehat{\delta} = E(\delta|\mathbf{r})$, for random effects.

## 4.3 REUCBHL ALGORITHM

We propose the ReUCBHL algorithm for contextual random effect SMAB, which is presented in Table 2. Fixed parameters $\theta$ and random effects $\delta$ are estimated by maximizing the h-likelihood. In round $t$, the algorithm chooses an arm $A_t = argmax_{k \in [K]} \mathbf{a}^T \mathbf{U}_{k,t}$ where $\mathbf{a} = (1/m)\mathbf{1}_m$ and $\mathbf{U}_{k,t}$ is the $m$-dimensional column vector of upper confidence bound which is given by

$$\mathbf{U}_{k,t} = \mathbf{X}_{k,t}\widehat{\beta}_t + \mathbf{Z}_{k,t}\widehat{\delta}_k + \widehat{\mathbf{c}}_{k,t} = \widehat{\mu}_{k,t} + \widehat{\mathbf{c}}_{k,t}, \tag{7}$$

where $\widehat{\mathbf{c}}_{k,t}$ is the $m$-dimensional column vector whose $j$th element is the $\sqrt{a\widehat{\tau}_{k,t,j}^2 \log(t)}$, $a$ is the tuning parameter, $\widehat{\tau}_{k,t,j}^2 = \widehat{var}(\widehat{\mu}_{k,t,j})$ and $\widehat{\mu}_{k,t,j}$ is the $j$th element of $\widehat{\mu}_{k,t}$.

Similar to univariate SMAB, in this study we define regret for multi-dimensional SMAB. Given the selected arm $k$, its random effect is realized as fixed $\delta_{0k}$. Lee et al. (2024) used the classical regret based on marginal model $\mathbf{X}_{k,t}\beta$ without accounting for arm effect $\delta_{0k}$. The optimal arm in the $t$th round is defined as the arm having $A_t^* = \arg\max_{k \in [K]}(\mathbf{a}^T \mathbf{X}_{k,t}\beta + \mathbf{a}^T \mathbf{Z}_{k,t}\delta_{0k})$ given contexts $\mathbf{X}_{k,t}$ and $\mathbf{Z}_{k,t}$. Then, the total regret in $n$ rounds of play is defined by

$$R_n = E\left\{\sum_{t=1}^n \left(\mu_{A_t^*,t} - \mu_{A_t,t}\right)\right\},$$

where $\mu_{A_t^*,t} = \max_{k \in [K]}(\mathbf{a}^T \mathbf{X}_{k,t}\beta + \mathbf{a}^T \mathbf{Z}_{k,t}\delta_{0k})$ is a fixed unknown constant given contexts $(\mathbf{X}_{k,t}, \mathbf{Z}_{k,t})$, $A_t = \arg\max_{k \in [K]}(\mathbf{U}_{k,t})$ in (7). Difference between random effect SMAB (3) and

| | |
|---|---|
| 1: | **INPUT:** number of random exploration rounds $d$ and tuning parameter $a$. |
| 2: | **for** each round $t \le d$ **do** |
| 3: | Sample an arm $A_t \in [K]$ randomly and observe $\mathbf{r}_{k,A_t}$. |
| 4: | **end for** |
| 5: | Calculate $\widehat{\theta}_d$. |
| 6: | **for** $t > d$ **do** |
| 7: | Observe the contexts $\{\mathbf{X}_{t,k}\}, \{\mathbf{Z}_{t,k}\}$ and compute $\widehat{\theta}_t \in H_t$ |
| 8: | Compute $U_{k,t}$ for each arm $k$ using equation (6) |
| 9: | Pull the arm $A_t = \mathbf{argmax_{k \in [K]}} a^T \mathbf{U}_{k,t}$ and observe $\mathbf{r}_{k,A_t}$ |
| 10: | **end for** |

Table 2: ReUCBHL algorithm for multivariate SMAB

contextual random effect SMAB (6) is that $\mu_{A_t^*,t}$ changes over time $t$ in the contextual model. We develop $n$ round regret bound for ReUCBHL algorithm in Table 2.

**Theorem 2.** *Under the multi-dimensional contextual SMAB (6) with arm-dependent noise, the $n$-round regret of ReUCBHL is*

$$R_n \le \sigma_{\max} \xi (\sqrt{2d(p+q+K)n \log(n)} + K\sqrt{2/\pi}),$$

where $\sigma_{\max}^2 = \max_{k \in [K]} \sigma_k^2$, $\xi = \max_{k \in [K], t \in [n]} \sqrt{\mathbf{a}^T (\mathbf{X}_{k,t} \mathbf{X}_{k,t}^T + \mathbf{Z}_{k,t} \mathbf{\Sigma}_0 \mathbf{Z}_{k,t}^T) \mathbf{a} / \sigma_k^2}$ and $d = \log(1 + \xi^2 nm/(p+q+K))/\log(1 + \xi^2/m)$.

## 4.4 RELATED WORKS

There are several variants of SMABs that allow for multi-dimensional reward such as combinatorial SMAB (Chen et al., 2013; Qin et al., 2014; Li et al., 2016) and multi-objective SMAB (Drugan & Nowe, 2013). However, they did not introduce the correlation structure. It was Lee et al. (2024), who introduced the correlation structure using the random effect model.

- C2UCB (Qin et al., 2014): Chen et al. (2013) introduced combinatorial UCB algorithm for analyzing the regret performance. Qin et al. (2014) proposed C2UCB (contextual combinatorial UCB) algorithm for contextual SMABs by extending the work of Chen et al. (2013).

- ME-CUCB (Lee et al., 2024): Lee et al. (2024) used weighted least squares estimators for fixed effects $\beta$, expectation-maximization algorithm for variance parameters and BLUPs for $\delta$.

## 4.5 NUMERICAL STUDIES FOR RANDOM INTERCEPT MODEL

For multivariate reward model (6), we consider numerical study when $\mathbf{X}_{k,t} = \mathbf{1}_m$ and $\mathbf{Z}_{k,t} = \mathbf{I}_m$ with $m = 10$, $K = 100$ and $n = 1,000$, where entries in $\mathbf{X}_{k,t}$ and $\mathbf{Z}_{k,t}$ do not depend on $t$. By setting $\beta = 1$ and $\mathbf{\Sigma}_0 = \mathbf{I}_m$, we compare the ReUCBHL algorithm with C2UCB (Qin et al., 2014) and ME-CUCB (Lee et al., 2024) algorithms in terms cumulative regrets over $n$ rounds of play. Each simulation experiment is averaged over 200 independent runs.

i) Arm-independent case: We assume that the reward noise $\sigma_k^2 = 1$ is constant for all arms $k$. Figure 2(a) shows a plot of average of $m$ cumulative regrets as a function of time horizon. In this case, ReUCBHL and ME-CUCB perform almost identically and perform better than C2UCB.

ii) Arm-dependent case: We assume $\sigma_k^2 = \log(k+1)$. The regret performance of the algorithms is shown in Figure 2(b). We observe that ReUCBHL outperforms the other algorithms. Arm-dependent assumption enhances the performance of ME-CUCB algorithm. The ReUCBHL performs the best in the multi-dimensional contextual SMABs.

## 4.6 Numerical studies for random slope model

Following Lee et al. (2024), we consider the $p = 10$-dimensional context matrix $\mathbf{X}_{k,t}$ whose the $j$th row columns $\mathbf{x}_{k,t}^{(j)}$ are generated by $\mathcal{N}(0, \mathbf{I}_p)$ and random coefficient model $\mathbf{Z}_{k,t} = \mathbf{X}_{k,t} \otimes \mathbf{1}_m^T$ that changes over time $t$ as $\mathbf{X}_{k,t}$. By setting $\beta = \mathbf{1}$ and $\mathbf{\Sigma}_0 = \mathbf{I}_{pm}$ with $m = 10$, $K = 100$ and $n = 1,000$, we compare the performance of the proposed ReUCBHL algorithm with C2UCB (Qin et al., 2014) and ME-CUCB (Lee et al., 2024) algorithms in terms cumulative regrets over $n$ rounds of play. Each simulation experiment is averaged over 200 independent runs.

i) Arm-independent case: We assume that the reward noise $\sigma_k^2 = 1$ is constant for all arms $k$. Figure 2(c) shows a plot of average of $m$ cumulative regrets as a function of time horizon. Here ReUCBHL and ME-CUCB perform almost identical and are better than C2UCB.

ii) Arm-dependent case: Consider $\sigma_k^2 = \log(k+1)$. The regret performance of the algorithms is shown in Figure 2(d). Again ReUCBHL outperforms the rest. Thus, the ReUCBHL algorithm is strongly recommended for multivariate contextual SMABs.

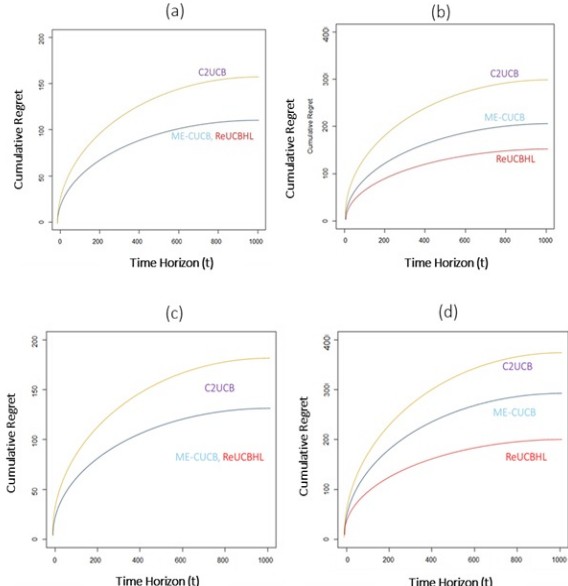

Figure 2: Comparison of average of cumulative regrets (a) arm-independent, (b) arm-dependent noise for the intercept-only model and (a) homogeneous, (b) arm-dependent noise for the intercept-only model for the model under the multivariate contextual random effect SMAB.

## 5 Conclusion

The UCB algorithm remains a cornerstone in reinforcement learning. Recently, random-effect bandit models have been introduced to exploit correlation structure across arms. However, existing works have focused on the SMABs with arm-independent noise variances. In practice, arm-dependent noise is ubiquitous and developing efficient algorithms under such setting is challenging due to difficulties in parameter estimation. In this study, we develop ReUCB algorithm for random effect SMABs with arm-dependent noises, which is easily implementable by simply minimizing the loss function (negative of h-likelihood) and computationally as fast as other algorithms. Furthermore, our experimental studies show that it outperforms all the existing state-of-art algorithms for SMABs and multivariate contextual SMABs. We should always use ReUCBHL algorithm because there is no loss in assuming arm-dependent noise as the regret is almost identical to the best algorithm in arm-independent cases. In arm-dependent cases, arm-independent random effect SMAB could be worse than TS. Throughout the studies, arm-dependent random effect SMAB always outperforms all the existing state-of-arts algorithms.

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

## APPENDIX A

Under the model (2), the (log-)h-likelihood (Lee & Nelder, 1996) of the fixed parameters $\theta = (\mu_0, \sigma_0^2, \sigma_1^2, ..., \sigma_K^2)^T$ and the random effect $\delta$ for the observed reward $\mathbf{r}$ in the $n$ round of play is defined by

$$h(\theta, \delta; \mathbf{r}) = -\frac{1}{2}(\mathbf{r} - \mu_0 \mathbf{1} - \mathbf{Z}\delta)^T \mathbf{\Sigma}^{-1}(\mathbf{r} - \mu_0 \mathbf{1} - \mathbf{Z}\delta) - \frac{1}{2}\log(2\pi\mathbf{\Sigma}) - \frac{1}{2}\delta^T \mathbf{D}^{-1}\delta - \frac{1}{2}\log(2\pi\mathbf{D}).$$

The marginal-(log-)likelihood $\ell$ can be obtained form $h$ by integrating out the random effect,

$$\ell(\theta; \mathbf{r}) = \log \int \exp\{h(\theta, \delta; \mathbf{r})\}d\delta.$$

Optimization of the joint likelihood $h(\theta, \delta; \mathbf{r})$ gives MLEs, maximizing $\ell(\theta; \mathbf{r})$ for $\theta$, and the BLUPs for $\delta$. However, it cannot give MLEs for the variances $(\sigma_0^2, \sigma_1^2, ..., \sigma_K^2)$. Recently, Lee & Lee (2023) suggested to use the new h-likelihood based on the canonical scale of random effects $\delta^c = \mathbf{A}^{1/2}\delta$,

$$h(\theta, \delta^{\mathbf{c}}; \mathbf{r}) = \log f(\mathbf{r}|\delta^c) + \log f(\mathbf{c}) = \ell(\theta) + \log f(\delta^c|\mathbf{r}).$$

Following Lee & Lee (2023), since $\delta|\mathbf{r}$ has the multivariate normal distribution

$$\delta|\mathbf{r} \sim N(\mathbf{A}^{-1}\mathbf{Z}^T\boldsymbol{\Sigma}^{-1}(\mathbf{r} - \mu_0\mathbf{1}), \ \mathbf{A}^{-1}),$$

we have

$$\delta^c|\mathbf{r} \sim N(\mathbf{A}^{-1/2}\mathbf{Z}^T\boldsymbol{\Sigma}^{-1}(\mathbf{r} - \mu_0\mathbf{1}), \ \mathbf{I}_K),$$

which leads to $\widetilde{\delta}^c = \mathbf{A}^{-1/2}\mathbf{Z}^T\boldsymbol{\Sigma}^{-1}(\mathbf{r} - \mu_0\mathbf{1})$. Thus, the resulting predictive likelihood becomes constant

$$\log f(\widetilde{\delta}^c|r) = -\frac{1}{2}\log|2\pi I_K| = -\frac{K}{2}\log(2\pi).$$

Thus, $\delta^c = \mathbf{A}^{1/2}\delta$ is the canonical scale to give the h-likelihood,

$$
\begin{aligned}
h(\theta, \delta; \mathbf{r}) \ = \ & -\frac{1}{2}(\mathbf{y} - \mu_0\mathbf{1} - \mathbf{Z}\mathbf{A}^{-1/2}\delta^c)^T\boldsymbol{\Sigma}^{-1}(\mathbf{y} - \mu_0\mathbf{1} - \mathbf{Z}\mathbf{A}^{-1/2}\delta^c) - \frac{1}{2}\log(2\pi\boldsymbol{\Sigma}) \\
& -\frac{1}{2}\delta^{cT}\mathbf{A}^{-1/2}\mathbf{D}^{-1}\mathbf{A}^{-1/2}\delta^c - \frac{1}{2}\log(2\pi\mathbf{A}^{1/2}\mathbf{D}\mathbf{A}^{1/2}),
\end{aligned}
$$

whose joint maximization gives the MLEs for the whole parameters $\theta$ and BLUPs for the random effect $\delta$.

## APPENDIX B

**Proof of Theorem 1.**

We derive regret bound given fixed parameters $\theta = (\mu_0, \sigma_0^2, \sigma_1^2, ..., \sigma_K^2)^T$ for the random effect SMAB (3). At time $t$, arm $k$ is pulled $n_{k,t} \geq 1$ times with sample mean $\bar{r}_k = \sum_{t=1}^{n_{k,t}} r_{k,t}/n_{k,t}$ and variance $d_{k,t} = \sigma_k^2/n_{k,t}$. The h-likelihood estimator of $\mu_k$ is

$$\widehat{\mu}_{k,t} = w_{k,t}\bar{r}_{k,t} + (1 - w_{k,t})\mu_0,$$

where $w_{k,t} = \sigma_0^2/(\sigma_0^2 + d_{k,t})$. Then, we have

$$\widehat{\mu}_{k,t} - \mu_k = S_{k,t} + B_{k,t},$$

where $S_{k,t} = w_{k,t}(\bar{r}_{k,t} - \mu_k)$ and $B_{k,t} = -(1 - w_{k,t})\delta_{0k}$. So, $S_{k,t} \sim N(0, \tau_{k,t}^2)$ with

$$\tau_{k,t}^2 = w_{k,t}^2 d_{k,t} = \frac{\sigma_0^4\sigma_k^2 n_{k,t}}{(\sigma_0^2 n_{k,t} + \sigma_k^2)^2}.$$

For any $x > 0$, the tail-probability is

$$\Pr\left(|\widehat{\mu}_{k,t} - \mu_k - B_{k,t}| \geq x\right) \leq 2\exp\left(-x^2/2\tau_{k,t}^2\right).$$

Set the confidence radii of arm $k$ in round $t$ as

$$c_{k,t} = \tau_{k,t}\sqrt{a\log t} = \tau_{k,t}\sqrt{2\log(1/\alpha_t)}$$

where $\alpha_t = t^{-a/2}$ and $a$ is the tuning parameter. Taking $a = 2$, we have

$$\Pr\left(|\widehat{\mu}_{k,t} - \mu_k - B_{k,t}| \geq c_{k,t}\right) \leq 2t^{-1}.$$

Define the global event

$$G_t = \bigcap_{k=1}^{K}\{|\widehat{\mu}_{k,t} - \mu_k - B_{k,t}| \leq c_{k,t}\}.$$

This ensures that confidence intervals hold for both the played arm and the oracle arm simultaneously. Let $G_t^c$ denote the complement of $G_t$. Then,

$$
\begin{aligned}
\Pr(G_t^c) &\leq 2Kt^{-1}, \\
\sum_{t=1}^{n} \Pr(G_t^c) &\leq 2K(1 + \log n).
\end{aligned}
$$

At round $t$, ReUCBHL chooses $A_t = \arg\max_k(\widehat{\mu}_{k,t} + \widehat{c}_{k,t})$. For the optimal arm $A^* = \arg\max_k \mu_k$, we have

$$
\mu_{A^*} - \mu_{A_t} = (\mu_{A^*} - \widehat{\mu}_{A_t,t} - \widehat{c}_{A_t,t}) + (\widehat{\mu}_{A_t,t} + \widehat{c}_{A_t,t} - \mu_{A_t}).
$$

On $G_t$,

$$
\begin{aligned}
\mu_{A^*} - \widehat{\mu}_{A_t,t} - c_{A_t,t} &\leq -B_{A^*,t}, \\
\widehat{\mu}_{A_t,t} + c_{A_t,} - \mu_{A_t} &\leq 2c_{A_t,t} + B_{A_t,t}.
\end{aligned}
$$

Thus,

$$
\mu_{A^*} - \mu_{A_t} \leq 2c_{A_t,t} + (B_{A_t,t} - B_{A^*,t}).
$$

On $G_t^c$ we use the trivial bound $\mu_{A^*} - \mu_{A_t,t} \leq \Delta_{\max}$, where $\Delta_{\max}$ is the maximum gap. We assume that the rewards $r_{k,t}$ are bounded in [0,1]. Taking expectations and summing,

$$
\begin{aligned}
R_n &= \sum_{t=1}^{n} E(\mu_{A^*} - \mu_{A_t}) \\
&\leq 2\sum_{t=1}^{n} E(c_{A_t,t}) + \sum_{t=1}^{n} E(B_{A_t,t} - B_{A^*,t}) + \Delta_{\max} \sum_{t=1}^{n} \Pr(G_t^c).
\end{aligned}
$$

For all $k, t$,

$$
\tau_{k,t}^2 = \frac{\sigma_0^4 \sigma_k^2 n_{k,t}}{(\sigma_0^2 n_{k,t} + \sigma_k^2)^2} \leq \frac{\sigma_0^2 \sigma_k^2}{\sigma_0^2 n_{k,t} + \sigma_k^2}.
$$

Therefore,

$$
c_{k,t} \leq \sqrt{2\log n} \sqrt{\frac{\sigma_0^2 \sigma_k^2}{\sigma_0^2 n_{k,t} + \sigma_k^2}}.
$$

Grouping by pulls of arm $k$ and bounding by an integral,

$$
\sum_{s=1}^{T_k(n)} c_{k,s} \leq \sqrt{8\log n} \frac{\sigma_k}{\sigma_0}(\sqrt{\sigma_0^2 T_k(n) + \sigma_k^2} - \sigma_k) + O(\sqrt{\log n}),
$$

where $T_k(n)$ is the total number of pulls of arm $k$. Sum across arms and apply Cauchy–Schwarz inequality,

$$
\sum_{t=1}^{n} c_{A_t,t} \leq \sqrt{8\log n} \frac{\sqrt{\sum_{k=1}^{K} \sigma_k^2}}{\sigma_0} \sqrt{\sigma_0^2 n + \sum_{k=1}^{K} \sigma_k^2} + O(K\sqrt{\log n}).
$$

Since $B_{k,t} = -(1 - w_{k,t})\delta_k = -\frac{d_{k,t}}{\sigma_0^2 + d_{k,t}}\delta_k$,

$$
\sum_{s=1}^{T_k(n)} |B_{k,s}| \leq |\delta_k| \sum_{s=1}^{T_k(n)} \frac{\sigma_k^2}{\sigma_0^2 s + \sigma_k^2} = O(|\delta_k| + |\delta_k|\frac{\sigma_k^2}{\sigma_0^2} \log T_k(n)).
$$

Thus,

$$
|\sum_{t=1}^{n}(B_{A_t,t} - B_{A^*,t})| \leq O(\delta_{\max} \frac{\sum_{k=1}^{K} \sigma_k^2}{\sigma_0^2} \log n),
$$

with $\delta_{\max} = \max_k |\delta_{0k}|$. For any fixed realization $\delta_{\mathbf{0}} = (\delta_{01}, \ldots, \delta_{0K})$,

$$R_n \leq C\sqrt{\log n} \, \frac{\sqrt{\sum_{k=1}^K \sigma_k^2}}{\sigma_0} \sqrt{\sigma_0^2 n + \sum_{k=1}^K \sigma_k^2} + O(\delta_{\max} \sum_{k=1}^K (\sigma_k^2/\sigma_0^2) \log n) + O(K\sqrt{\log n}),$$

where, constant $C = 4\sqrt{2}$. This completes the proof.

**Proof of Theorem 2.**

The model (6) can be written by

$$r_{k,t}^* = \mu_{k,t}^* + e_{k,t}^*$$

with $r_{k,t}^* = \mathbf{a}^T \mathbf{r}_{k,t}/\sigma_k$, $\mu_{k,t}^* = \mathbf{a}^T \mu_{k,t}/\sigma_k$ and $e_{k,t}^* = \mathbf{a}^T \mathbf{e}_{k,t}/\sigma_k \sim \mathcal{N}(0, 1/m)$. Let $\sigma_{\max}^2 = \max_{k \in [K]}\{\sigma_k^2\}$. Because $\sigma_*/\sigma_k \geq 1$ for all $k$, then the $n$-round regret can be obtained by

$$R_n \leq E\left\{\sum_{t=1}^n \left(\frac{\sigma_{\max}}{\sigma_k}\mu_{A_t^*} - \frac{\sigma_{\max}}{\sigma_k}\mu_{A_t,t}\right)\right\} = \sigma_{\max} R_n^*.$$

where $R_n^* = E\left\{\sum_{t=1}^n \left(\frac{1}{\sigma_k}\mu_{*,t} - \frac{1}{\sigma_k}\mu_{A_t,t}\right)\right\}$. When we take expectation over $\mu_{A_t,t}$, $R_n^*$ becomes the Bayes regret. For simplicity of calculation, we use Bayes regret to analyze regret bound. Following **Theorem 1** of Zhu & Kveton (2022b), we have

$$R_n^* \leq \xi\sqrt{2d(p+q+K)n\log(n)} + \sqrt{2/\pi}\xi K.$$

This completes the proof.

