# OpenReview forum: "Random Effect Bandits using h-Likelihood"
_ICLR.cc/2026/Conference — ICLR 2026 Conference Withdrawn Submission_

### Official Review · Reviewer_Hp8H · 2025-10-20

**Soundness:** 2
**Presentation:** 3
**Contribution:** 2
**Rating:** 2
**Confidence:** 4

**Summary:**

This work investigates random effect bandits where each arm's expected reward is sampled i.i.d from the distribution $N(\mu_0, \sigma_0^2)$. The paper's novel contribution is the consideration of arm-dependent noise, and achieves a variance-dependent regret guarantee. Simulation results validate the efficiency of the proposed method.

**Strengths:**

1. The paper successfully derives a variance-dependent regret guarantee, achieved by explicitly modeling arm variance.

2. Simulation results confirm the proposed algorithm's effective performance.

**Weaknesses:**

1. Though the author provides a variance-dependent regret guarantee, there lacks a crucial comparison with the results from the more general stochastic multi-arm bandit (MAB) problem. In fact, Random Effect Bandits is a sub-class of the stochastic MAB problem, as the expected reward for all arms must be sampled from the same distribution $N(\mu_0, \sigma_0^2)$, whereas in the general MAB setting, each arm can have an arbitrary expected reward $\mu_k$.

It is generally expected that the regret is much smaller when the problem setting is more restricted. However, it seems that the classic analysis for the instance-dependent regret for the multi-armed bandit problem can directly yield an improved result. (See Theorem 8.1 in the classic textbook for bandit theory by Lattimore and Szepesvári [1]).

[1] Lattimore, Tor, and Csaba Szepesvári. Bandit algorithms. Cambridge University Press, 2020.

In detail, when all expected rewards are sampled from $N(\mu_0, \sigma_0^2)$, with high probability, the gap between the optimal reward and the sub-optimal reward is at least $\Omega(\sigma/\sqrt{\log K})$. Under this situation, Theorem 8.1 directly implies an $\mathcal{O}(\log T)$ regret, while the ReUCBHL algorithm only achieves an $\mathcal{O}(\sqrt{T})$ regret.

In summary, this work proposes a new algorithm for a more restrictive setting, but the regret guarantee is worse than the result obtained by applying a classic bandit algorithm for a general environment to this specific setting. This highly impacts the theoretical contribution of this work.

2. On the other hand, the experiments only provide simulation results. Even though these results support the efficiency of the proposed algorithm, simulation results cannot individually support the contribution of this paper without validation on real-world or complex benchmark datasets.

**Questions:**

See Weakness.

---

### Official Review · Reviewer_6unJ · 2025-10-30

**Soundness:** 3
**Presentation:** 3
**Contribution:** 3
**Rating:** 6
**Confidence:** 3

**Summary:**

Recently, random effect stochastic multi-armed bandit (SMAB) has been proposed where reward feedback is modeled as random effect. However, it has not been well formulated yet in likelihood perspectives. Furthermore, individual noise variance can be arm-dependent. This paper proposes a novel random effect upper confidence bound (ReUCBHL) algorithm, based on h-likelihood. The likelihood approach is conceptually easy and can be implemented by simply minimizing the loss (negative h-likelihood). The algorithm can be applied to SMAB with univariate and multivariate rewards under arm-dependent noise variances. It can be further extended to contextual multivariate bandit. Theoretical justification and simulation studies demonstrate that ReUCBHL consistently achieves better regret performance compared to the baseline algorithms. These results highlight the effectiveness of the proposed algorithm.

**Strengths:**

1. The studied problem, stochastic multi-armed bandit (SMAB), is a fundamental problem in online learning, and has various applications such as clinical trials, recommendation systems and robotics.
2. This paper proposes to formulate random effect SMAB from the likelihood perspective and consider arm-dependent noises.
3. This paper designs a novel random effect upper confidence bound (ReUCBHL) algorithm, based on h-likelihood, and extends it to the multivariate SMAB setting. Both regret guarantees and empirical evaluations are provided.

**Weaknesses:**

1. The authors should discuss more on the motivation of the proposed random effect SMAB formulation by connecting it with real-world applications, especially when compared to existing random effect SMAB formulations. For example, what are the advantages of considering arm-dependent noises?
2. Can UCB, BUCB and TS be applied to the proposed random effect SMAB problem with arm-dependent noises? In other words, is the comparison in empirical evaluations a fair comparison? The authors should discuss more on this point.
3. Minor comment: The format of the algorithm pseudo-codes can be improved.

**Questions:**

Please see the weaknesses above.

---

### Official Review · Reviewer_gcQm · 2025-10-31

**Soundness:** 1
**Presentation:** 1
**Contribution:** 1
**Rating:** 0
**Confidence:** 4

**Summary:**

The paper studies two variants of MABs, the random effect SMAB with arm-dependent noise and multi-dimensional contextual random effect SMAB with arm-dependent noise. Extremely simple methods, maximizing the corresponding h-likelihood functions, are introduced to select

**Strengths:**

I couldn't find strengths in this submission. Please see the Weaknesses part.

**Weaknesses:**

- The paper did not justify the necessity and the challenges of the new problem settings and their importance. Formulations (3) and (6) appear more structural than the conventional settings, hence making them easier problems.

- The paper is poorly written
  - Equations (2) vs (3).
  - Section titles for Section 2.3 and Section 3.
  - Notations \Sigma and D_K are not defined.
  - Disconnection from the main algorithm (Table 1) and the key notion h-likelihood (line147).
  - Duplicate “The simple joint maximization …” (lines 150 and 355) without further justifications.
  - The forms of Theorems 1 and 2 differ drastically, but the formulations (3) and (6) are close. There is no discussion of the difference.
  - The curves in Figures 1 and 2 do not appear to be experimental outcomes.
  - Conflicting claims: line 247 vs line 482.

- There is no LLM Usage section in this paper, but the paper contains several mistakes that a human will not make.
  - Reversed names in two references: Shipra Aggarwal and Goyal Navin
  - In consistent names in the link of the MovieLens dataset
  - The Kyungbok Lee et. al paper published in AAAI cannot be PMLR.
  - The reading experience is locally coherent, but many mismatches, even conflicts, globally. (This can be too objective.)

**Questions:**

N/A.

---

### Official Review · Reviewer_xSg4 · 2025-11-01

**Soundness:** 2
**Presentation:** 2
**Contribution:** 2
**Rating:** 2
**Confidence:** 3

**Summary:**

This paper introduces the h-likehood approach to derive UCB for stochastic multi-armed bandits with arm-dependent noise model. The proposed algorithm, ReUCBHL, enjoys the same regret bounds as prior baselines, and outperform the baselines when the arm noises are arm-dependent.

**Strengths:**

1. Better empirical performance: The proposed algorithm enjoys better empirical performance.
2. New Statistical Technique: The paper introduces the h-likelihood, an MLE method considering the arm-dependent noise, to construct the UCB index.

**Weaknesses:**

1. The writing of this paper is hard to follow. In the intro’s last paragraph, the author mentioned h-likelihood multiple times, but the definition or its intuition is not explained.
    - In Table 1, Line 2, the $k$ should be $K$.
    - Line 205, it is not exactly MLE that UCB uses. It uses concentration inequalities.
    - Table 2, Line 8, should it be equation (7)?
2. Lack of novelty: Although the algorithm performs better, it relies on a strict Gaussian parametric assumption (both the reward and the variance of the noise). Therefore, the new technique, h-likelihood, for UCB construction seems to be an application of the MLE together with the property of the Gaussian distribution itself. If that is true, then the techniques used in this paper are basic and difficult to extend to more general settings.
    1. What’s the difference between h-likelihood and the likelihood calculated by MLE?
3. It would be better to derive the instance-dependent bounds, as is a common practice for applying UCB-like algorithms to stochastic bandits.
4. The proof of Theorem 2 is informal, and the regret definition used in the proof is different from that in the main paper (Section 4.3).

**Questions:**

See weaknesses.

---

### Note · Authors · 2025-11-13

I have read and agree with the venue's withdrawal policy on behalf of myself and my co-authors.